# Interfacial Properties and Melt Processability of Cellulose Acetate Propionate Composites by Melt Blending of Biofillers

**DOI:** 10.3390/polym14204286

**Published:** 2022-10-12

**Authors:** Ji-Eun Lee, Seung-Bo Shim, Jae-Hyung Park, Ildoo Chung

**Affiliations:** 1Department of Polymer Science and Engineering, Pusan National University, Busan 46241, Korea; 2Korea Institute of Footwear & Leather Technology, 152 Dangamseo-ro, Busanjin-gu, Busan 47154, Korea

**Keywords:** interfacial properties, melt blend cellulose acetate propionate, biocomposite, automobile interior parts

## Abstract

A series of eco-friendly biocomposites with improved mechanical properties and interfacial interaction were prepared by melt-mixing natural fibers using a cellulose acetate derivative as a polymer matrix and used to evaluate their mechanical, thermal, and morphological properties. The natural fiber used as a biofiller was pre-surface-treated by a refining process using alkali and natural enzymes to improve compatibility and increase interfacial bonding with biopolymer substrate. To increase the processability of the cellulose material, the raw material was plasticized and the composition prepared in the form of pellets in a twin-screw extruder by mixing with an additive before being molded through an injection process. For each composition, the interfacial bonding force between different materials was confirmed through morphology analysis and evaluation of mechanical and thermal properties. When biofillers and a viscosity modifier were used at the same time, the fabricated biocomposites had controllable crystallinity, stiffness, and elasticity and showed improved mechanical strength, such as tensile strength and flexural strength. These results indicated that interfacial properties could be increased through interfacial interactions between two different components due to appropriate surface treatment. In addition, it was confirmed that a composition having interfacial interaction, not a simple mixture, could be prepared by lowering both glass transition and melting temperature. The lowering of glass transition temperature increased the elasticity of the biocomposites, which have the potential advantage of easier melt processing when applied to various injection parts.

## 1. Introduction

Currently, for the purpose of strengthening regulations and light weight, the proportion of plastic in automobile interior parts is rapidly increasing. This is also due to requirements for automotive interior and exterior parts that are complex, eco-friendly, and highly sensitive [1,2,3,4,5]. Figure 1 shows the eco-friendly and lightweight trends pursued by global automobile brands.

In particular, natural fiber-reinforced plastic (NFRP) parts are reported to be more advantageous than glass fiber-reinforced parts after analyzing environmental impact, damage to a product during its life cycle, and the depletion of petroleum-based resources. As the warming problem caused by carbon dioxide emissions and environmental ecological pollution due to intensified microplastics intensifies, interest in developing eco-friendly materials that can replace existing hard-to-decompose petroleum-based plastics is growing.

Figure 2 shows the natural cycle life according to the use of biocomposites. Among eco-friendly materials, natural fiber-reinforced polymer composites, called biocomposites, are generally composed of natural fibers used as reinforcing fibers and non-biodegradable or biodegradable polymer used as a matrix [2,6,7]. The natural fiber-reinforced biocomposite is biodegradable after a certain period of time and recyclable. It can be applied as a sustainable composite. Accordingly, attempts are actively being made to overcome environmental problems related to plastics by adding engineering functionality from natural resources and developing raw materials with biodegradable properties. To find a solution, the development of biocomposite materials with reinforced mechanical strength is rapidly increasing.

In general, a strong interfacial adhesion between fibers and the matrix plays the most important role in obtaining excellent properties and performance with composite materials. This concept can also be applied to biocomposite materials using natural fibers [7,8,9,10]. As depicted in Figure 3, the surface of a hydrophilic natural fiber has poor interfacial bonding with the hydrophobic polymer matrix resin [11,12,13,14]. When used without modification, natural fiber used for manufacturing biocomposites has negative effects on water resistance, which may negatively impact the mechanical properties of biocomposites. Therefore, many studies have been conducted to improve the performance and physical properties of biocomposite materials by increasing the interfacial bonding force with polymer resin through modification of natural fibers [15,16,17].

Representative approaches for modifying the surface of natural fibers include changing fiber surface from hydrophilicity to hydrophobicity or reducing hydrophilicity, which is a method that introduces a chemical functional group to the fiber surface to promote mutual attraction between the resin and the fiber surface. There are also other methods, such as improving interfacial adhesion through mechanical bonding by increasing the roughness and surface area of the fiber, as well as a method which improves bonding strength through alkali treatment or simple washing to remove hemicellulose, pectin, wax components, or surface impurities present on the surface of fibers [18]. Alternatively, there is a method of introducing a third interphase to the interface between the natural fiber and the resin to improve the interfacial bonding force by mutual attraction of the fiber-interfacial phase-resin.

As shown in Figure 4, it is commonly accepted that natural cellulose has 70% crystalline and very good mechanical properties, but it is difficult to process due to the strong intermolecular bonding force and crystallinity from hydrogen bonding between the terminal OH groups. It is also difficult to mold because it decomposes before reaching the melting point [19,20].

For this reason, cellulose is converted into cellulose derivatives to prevent decomposition at the melting point and is processed by solution or the melt method. Cellulose derivatives are products obtained by oxidation, substitution, and other chemical treatment of wood-based materials, and are used as raw materials for various chemical industries depending on the kinds of functional groups substituted. Among various biomaterials, cellulose acetate (CA) is one of the most widely applied materials for automobile interior materials. CA has a high melting temperature because of its strong hydrogen bonding. Its melting temperature is not significantly different from its degradation temperature. Thus, decomposition may occur during its processing. To solve this problem, plasticizers such as dioctyl phthalate (DOP), dibutyl phthalate (DBP), diethyl phthalate (DEP), dibutyl sebacate (DBS), polyethylene glycol (PEG), glycerin (GC), and triacetin (TA) can be used. In particular, a CA plasticizing composition complexed with nano fillers is being studied frequently as a substitute for PP/TPO-based automobile parts. Figure 5 shows the structures of various cellulose derivatives.

In this study, by-products generated in the manufacturing process were reused as raw materials. By removing impurities on the surface through a scouring process, the compatibility with the polymer substrate was increased in order to improve interfacial adhesion and to improve physical properties. In addition, a change in properties according to composition ratio was observed by blending a biofiller for plastic additives [22], registered as Korea Patent Technology 10-1962239, without requiring a separate surface treatment. As a base material of the biocomposite, a cellulose ester-based resin capable of injection molding based on cellulose was used in this study. A plasticized CA composition was prepared by melt blending an eco-friendly plasticizer to lower the glass transition temperature and enable injection molding. CA substrate and additive were manufactured as a composite in a single process through melt blending using a twin-screw extruder. Mechanical properties of biocomposite materials manufactured by melt blending, such as tensile strength, flexural strength, melt behavior, and thermal properties, were then evaluated to establish the optimal composition formulation and to evaluate their applicability as automotive interior materials in parts such as door trims and garnishes, shown in Figure 6.

## 2. Materials and Methods

### 2.1. Materials 

As a biofiller, Kenaf fiber with an average length of 20 mm from Hanyang Materials (Seoul, Korea) was used in this study. After removing impurities through surface refining and atomization processes, Kenaf fiber products with a length of less than 5 mm were used. In addition, a bioadditive from Lignum Co., Ltd. (Daejeon, Korea, product name: SSEIF Biofiller), a brown powder product having a specific gravity of 1.2, a particle size of 20 μm or less, and a moisture content of 5% or less, was used without surface treatment [22]. Eastman’s cellulose acetate propionate (Kingsport, TN, USA, product name: CAP482-20) was used in this study. Cellulose acetate propionate (CAP) is an important thermoplastic cellulose ester. It is a tough and easy to process thermoplastic of high clarity with little to no odor. As a plasticizer, polyethylene glycol (PEG300) from Dae-Jung (Siheung, Korea) was used. Lactic acid from Aldrich (Milwaukee, WI, USA) was used as a viscosity regulator. In addition, adipic acid polyester (product name: OLI-20N) obtained from Aekyung Petrochemical Co., Ltd. (Seoul, Korea). was used as an auxiliary plasticizer without purification.

### 2.2. Surface Treatment of Natural Fibers

Since compatibility is reduced due to high hydrophobicity and impurities on the surface of natural fibers, impurities were removed from the surface through a process called refining, or scouring, as shown in Figure 7, to improve compatibility with the biopolymer substrate. In this study, alkali-based sodium hydroxide (NaOH) from Dae-Jung (Siheung, Korea) and natural enzyme α-amylase from Dong-A Petrochemical Co., Ltd. (Seoul, Korea) was used. By checking the properties and weight changes of natural fibers according to the content and composition of the refining agent, one product from which surface impurities were most effectively removed was selected. After removing impurities on the surface of natural fibers, reinforcing raw materials consisting of natural fibers of about 5 mm in length were obtained through a pulverization process. 

### 2.3. Preparation of Blending Composite and Plasticized Cellulose Acetate Propionate

Cellulose acetate propionate (CAP) was used after drying for 4 h in an oven at 80 °C. The dried CAP was mixed with a plasticizer (polyethylene glycol (PEG300)) by content and mixed for 24 h at a speed of 30 rpm in a hermetic mixer at about 20~25 °C. Lactic acid was added to the first plasticized CAP composition as a viscosity modifier and stirred for about 2 h to prepare a pellet-type composition through a melt blending process using a co-rotating twin-screw extruder [23]. Biofillers (natural fiber and SSEIF) were added simultaneously to the poly(lactic acid) input step. The extruder used in this experiment was a coaxial twin-screw extruder (L/D 40, screw diameter of 32 mm, 3 die holes) from EM Korea (Changwon, Korea). The twin-screw extruder die temperature was maintained at 190–200 °C and the discharge rate was 10 kg/hr. After cooling in a cooling bath, a rotary cutter was used to produce cylindrical pellets with a diameter of 1 to 2 mm and a length of 3 mm. The detailed process of melt blends is shown in Figure 8. The screw configuration of extruder is shown in Figure 9.

### 2.4. Preparation of Injection Specimen

To evaluate mechanical performance of the prepared composite composition, injection test pieces were manufactured under the following conditions: temperature of 200 to 220 °C, pressure of 1625 kg/cm^2^, injection time of 3 s, holding pressure of 140 bar, and a cooling time of 20 s. The test piece was manufactured using a Dongshin Hydraulic PR170HY (150 kgf/cm^2^ clamping force). It was then analyzed according to the standard test method for tensile properties of plastics (ASTM).

### 2.5. Characterization 

Mechanical properties of biocomposites were measured according to ASTM regulations using a universal testing machine (Instron M4465, Instron, Norwood, MA, USA, hereinafter UTM). A total of 5 specimens were used for measuring physical properties, and the average value was representatively described by measuring each value. Tensile strength of the composite material was measured at room temperature using Universal Test Machine equipment according to ASTM D638. A 50 kN cell was used for the load cell with a crosshead speed of 50 mm/min. Flexural strength was measured in the same equipment as tensile strength according to ASTM D790 with a speed of 30 mm/min. Impact strength was measured using an Izod impact tester with a notched specimen at room temperature in accordance with the standards of ASTM D256. Measurement of tensile strength and flexural strength is shown in Figure 10.

Thermal transition of blends was measured under nitrogen flow using a differential scanning calorimeter (DSC, TA instruments Q 100, New Castle, DE, USA) at a heating rate of 10 °C/min. Glass transition temperature (Tg) and melting temperature (Tm) were determined from the second heating scan. Morphologies of cross-sections of blends were studied by scanning electron microscopy (SEM, SEC SNE-3000MB, Seoul, South Korea) with an acceleration voltage of 30 kV at room temperature. 

## 3. Results and Discussion

### 3.1. Surface Treatment of Natural Fibers

Biorefining refers to a technique of discarding sodium hydroxide, the main drug of the conventional refining method, and refining it under mild conditions using enzymes or microorganisms. It is a technology that uses enzymes such as propeptidase, a type of peptidase, to decompose and solubilize pectin, which constitutes the primary wall of cellulose fibers, to remove all waxes, oils and fats, and contamination. Although the main body of the enzyme is a protein, it is often a complex protein in which sugar chains or chemical substances are bound to the protein. The enzymes that can be used for biorefining are basically any enzymes that decompose protopectin (water-insoluble pectin). In this study, we tried to maximize the effect of refining by mixing the existing refining method and the biorefining method. The weight reduction rate according to the refining time of natural fibers is shown in Table 1.

It is generally known that the composition of Kenaf fiber consists of 45–57% cellulose, 21.5% semi-cellulose, 8–13% lignin, and 3–5% pectin [24,25,26,27,28]. As a result of measurements obtained by the refining method used in this study, it is judged that the remaining impurities besides cellulose have been removed, as a weight reduction of about 48% was observed. These results are considered as proof that it is a very effective method for removing lignin, hemicellulose, and wax, with a short purification time of 6 h when the NaOH purification method and the natural enzyme α-amylase purification method are applied [29]. Compared to the purification method only using NaOH, the mixed method involving both NaOH and α-amylase could shorten the purification time and is considered to be very effective in reducing wastewater and volatile organic compounds (VOC) which could be generated by the residual cleaning solution when alkali refining agents are used.

Figure 11 and Figure 12 show the appearance of Kenaf fibers before and after refining and final grinding. At this time, since the refined natural fiber is a long fiber, fiber particles were pulverized to secure mixability and uniformity with the polymer when manufacturing the biocomposite material. The size of the final fiber product used as an additive was measured as having an average length of 5 mm. Grinding was performed twice to compare the texture and size of each product. In the case of Kenaf, as a result of the second grinding test, only particles were smaller. The feel after final grinding was judged to be similar to that of primary grinding, and the primary crushed product was used as an additive.

### 3.2. Biocomposite Structure

Composition characteristics of the prepared Kenaf fibers (after secondary grinding) and SSEIF biofillers (prepared through special surface treatment in the range of 0.5 to 3.0 phr through melt blending) were evaluated. Detailed formulations of each composition are shown in Table 2.

Figure 13 shows the appearance of each composition and the injection specimen. Figure 13 shows the interfacial compatibility and dispersion characteristics for the fracture surface of the composition prepared by melt blending biofillers. It can be judged that crystallinity decreased when the appearance of the plasticized composition became transparent. The difference in transparency and color depended on processing conditions and the mixing of the plasticizer. In general, the transparency of plasticized CA compositions is known to be greater than 90%. If the plasticizing effect and compatibility are poor, it is believed that a white opaque composition will be obtained instead of a transparent composition. Cellulose acetate alone has a problem in that melt processing is difficult due to its strong intermolecular hydrogen bonding and high glass transition temperature. Therefore, a plasticizing process using a compatible low molecular weight material is essential. However, when a single plasticizer is used, it has a miscibility threshold, which limits plasticization. Therefore, in this study, polyethylene glycol and adipic acid polyester were mixed and used. Results of morphology analysis confirmed excellent plasticity, even at a low content of 10 phr.

As a result of this experiment, it was confirmed that the fracture surface of the composition exhibited a very uniform shape when primary and secondary plasticizers and lactic acid, a viscosity modifier, were added to cellulose acetate propionate (Figure 14a,b). It was judged that interfacial separation did not occur due to excellent compatibility between the heterogeneous polymer matrix and the plasticizer. When natural fibers were used, some fibers were pulled out from the matrix. It was observed that the uniformity inside the polymer matrix decreased somewhat with an increase in the amount of additives. This might be because the interfacial bonding force between the bioadditive and the polymer matrix was somewhat weak. This trend was confirmed to be more severe in the case of biofillers than in surface-treated Kenaf fibers. In order to improve the poor interfacial interaction between the hydrophilic natural fibers and the hydrophobic polymer matrix, maleic anhydride (MAH) graft polymers are sometimes used as compatibilizers [30]. Therefore, in order to increase the interfacial compatibility between the polymer matrix and the natural fiber additive, it was determined that it was necessary to use a compatibilizer exhibiting polarity.

### 3.3. Fourier Transform Infrared Spectroscopy (FT-IR)

The FT-IR spectra of the composites, shown in Figure 15, show characteristic absorption peaks at 2900 (C–H stretch), 1750 (C=O stretch), and 1100–1050 cm^−1^ (C–O stretch), with concurrent disappearance of the OH stretching peak at 3650–3000 cm^−1^ due to the steric element. In particular, it was found that the intensity of the C–O stretching peak at 1100–1050 cm^−1^ increased with increasing amounts of biofiller.

### 3.4. Mechanical Properties of Biocomposites

Table 3 and Figure 16 list the mechanical strengths of the compositions. The blend of PEG reduced the glass transition temperature of CA and enhanced the tensile strength [31,32]. When lactic acid was used as a viscosity modifier, the flowability increased as the molecular weight decreased, mechanical strength slightly decreased. This was judged to be caused by a decrease in crystallinity inside the composite material. When impact strength was improved, the rigidity decreased but the elasticity increased. It was found that the tensile strength and flexural strength were increased with an increased amount of the biofiller, whereas the impact strength was significantly decreased. This seemed to be the opposite for viscosity modifiers. In the case of natural fibers surface-treated with alkali, hemicellulose from the surface was removed and the content of cellulose was increased and combined with the polymer matrix composed of cellulose. For this reason, it was judged that the tensile strength and flexural strength were improved by increasing stiffness according to the increased amount of the additive. On the other hand, as the elasticity decreased, the impact strength also decreased. 

### 3.5. Thermal Properties of Biocomposites (DSC and TGA Analysis)

Biofiller particles distributed in the polymer matrix can affect thermal properties. Filler can change the micromorphology of the polymer and act as a nucleating agent, which can thus affect the shape, size, and crystallinity of the crystal. On the other hand, the polymer layer present on the particle surface has reduced mobility and thus a decreased crystallization rate. In this regard, thermal properties of the CA composition were investigated. Results are shown in Table 4 and Figure 17. When CAP482-20 is used as a polymer matrix, it is known that the melting temperature is 188–210 °C and the glass transition temperature is 147 °C. For the plasticized CAP composite, the glass transition temperature was measured to be around −44 °C and the melting temperature around 170 °C. 

A more detailed analysis showed that the primary melting point increased and the secondary melting point decreased as the biofiller was added, and the enthalpy of melting showed a tendency to decrease. In general, in a polymer blend, the melting point of each component polymer changes according to the composition and compatibility degree of each component polymer, and the degree of the change indicates the degree of compatibility for the blend. The melting point lowering phenomenon occurs due to the dilution effect between the polymers when there is compatibility in the amorphous region, and the higher the compatibility, the more the melting point is lowered.

These results indicated that the molecular weight and crystallinity of the composition were reduced. The structural density, such as hydrogen bonding and covalent bonding, was also reduced due to the use of additives. However, there was an interaction between interfaces, rather than simply mixing.

Due to the reductions in glass transition temperature and melting temperature, the deterioration temperature and the gap of the raw material were widened, which had an advantage in that melt processing, such as extrusion and injection, became easier. Thus, it can be applied to various fields of parts. TGA(b) results showed that thermal decomposition started at about 370 °C. Thus, it could be judged that thermal stability was excellent.

## 4. Conclusions

In this paper, biocomposites were fabricated by melt blending plasticized CAP with pre-surface-treated natural fiber and the morphology, thermal properties, and mechanical strength of each composition was evaluated. An attempt was made to remove impurities on the surface through a refining process of natural fibers in order to increase compatibility with the polymer matrix. When biofillers and a viscosity modifier were used at the same time, the crystallinity, stiffness, and elasticity of the composite could be controlled, together with the improvement in mechanical properties such as tensile strength and flexural strength. In addition, unlike a simple mixture, it was confirmed that biocomposites with stronger interfacial interactions could be prepared by lowering glass transition temperature and melting temperature, leading to increased elasticity of composites. These results demonstrate an advantage in terms of easier melt processing which can be applied to various fields of injection parts. Since the biocomposite prepared in this work exhibited significantly higher mechanical strength, as compared to commercial glass fiber-reinforced plastic material used in the past, it has great potential for use in the production of automobile interior materials, electronic housings, and various household items. Future research will focus on quantitative studies including the degree of crystallinity. 

## Figures and Tables

**Figure 1 polymers-14-04286-f001:**
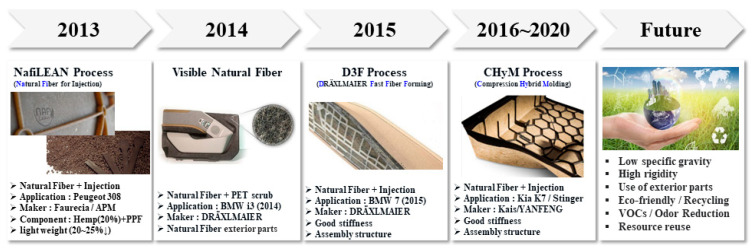
Eco-friendly and lightweight trend of global automobile parts.

**Figure 2 polymers-14-04286-f002:**
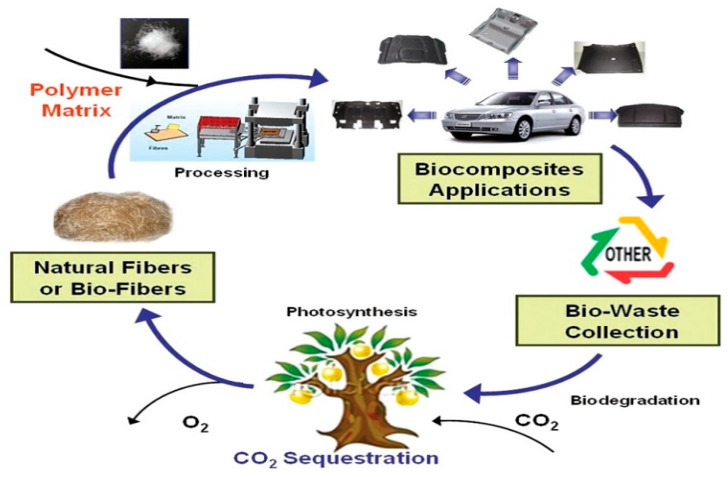
Natural life cycle of a natural fiber and biocomposite part [5].

**Figure 3 polymers-14-04286-f003:**
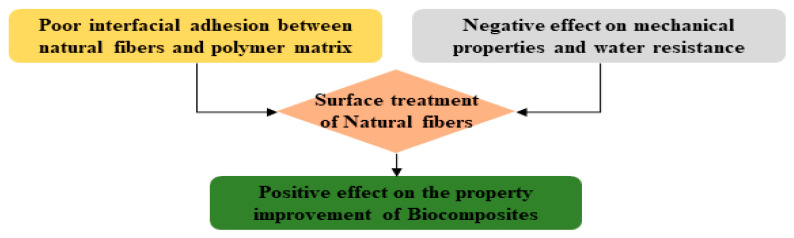
The need for surface treatment of natural fibers to improve the performance of biocomposites [14].

**Figure 4 polymers-14-04286-f004:**
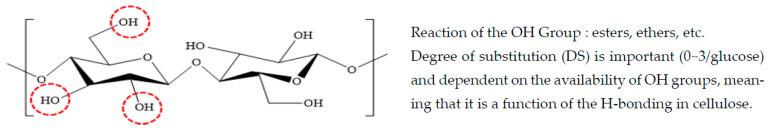
Structure of Cellulose.

**Figure 5 polymers-14-04286-f005:**
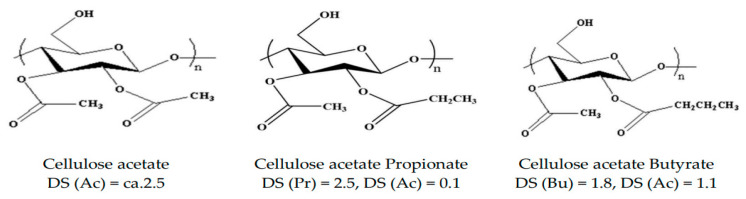
Representative structures of cellulose esters for plastics [21].

**Figure 6 polymers-14-04286-f006:**
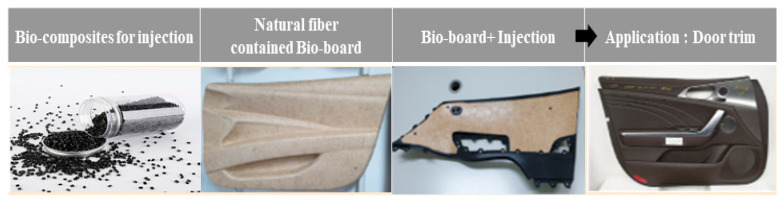
Application to biocomposite materials and automobile interior materials.

**Figure 7 polymers-14-04286-f007:**
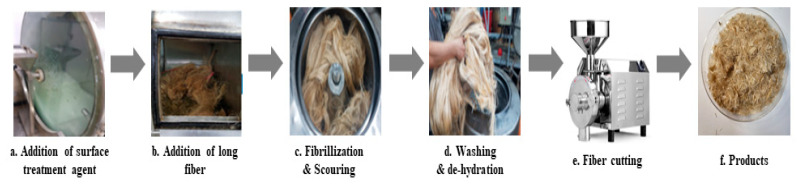
Process of surface treatment for natural fibers.

**Figure 8 polymers-14-04286-f008:**
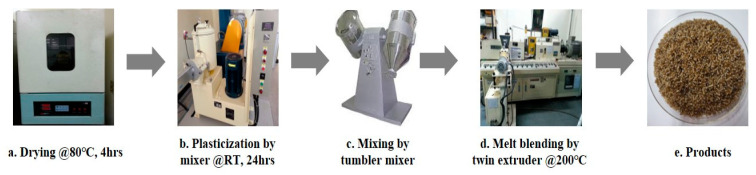
Process of preparing the cellulose acetate propionate–biofiller composite by melt blending.

**Figure 9 polymers-14-04286-f009:**
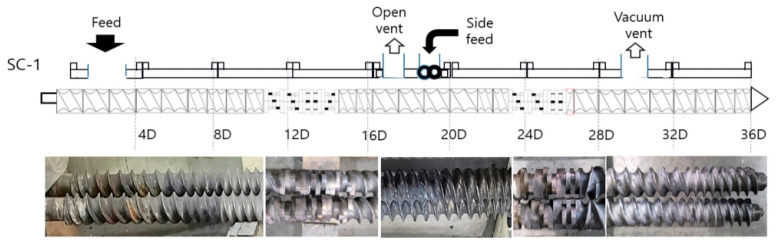
A general screw configuration of an intermeshing co-rotating twin-screw extruder for compounding polymeric materials.

**Figure 10 polymers-14-04286-f010:**
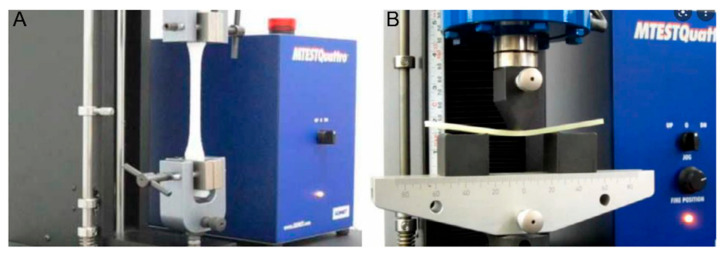
Comparison of the tensile strength (**A**) and flexural strength (**B**) measurement methods using a UTM.

**Figure 11 polymers-14-04286-f011:**
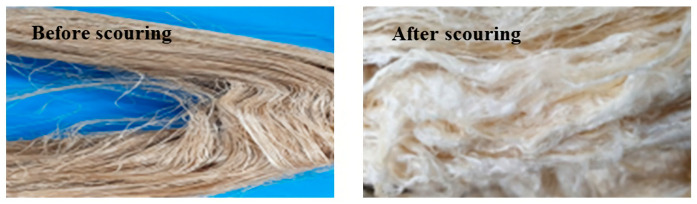
Before and after treatment of natural fiber.

**Figure 12 polymers-14-04286-f012:**
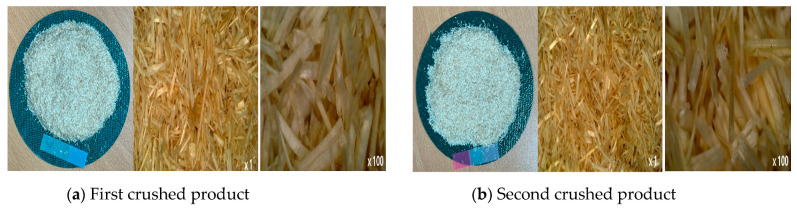
Images of cleaned and crushed Kenaf fibers after surface treatment.

**Figure 13 polymers-14-04286-f013:**
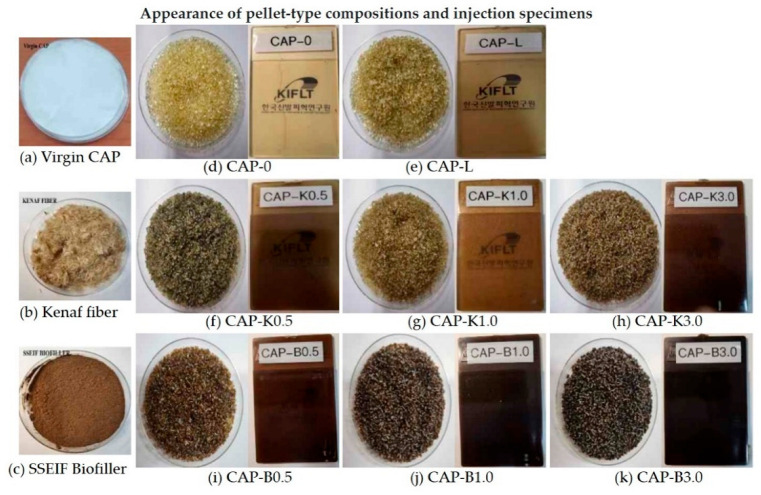
Appearance of the raw materials and biocomposite: (**a**)Virgin CAP; (**b**) Kenaf fiber; (**c**) SSEIF biofiller (**d**) CAP-0; (**e**) CAP-L; (**f**) CAP-K0.5; (**g**) CAP-K1.0; (**h**) CAP-K3.0; (**i**) CAP-B0.5; (**j**) CAP-B1.0; and (**k**) CAP-B3.0.

**Figure 14 polymers-14-04286-f014:**
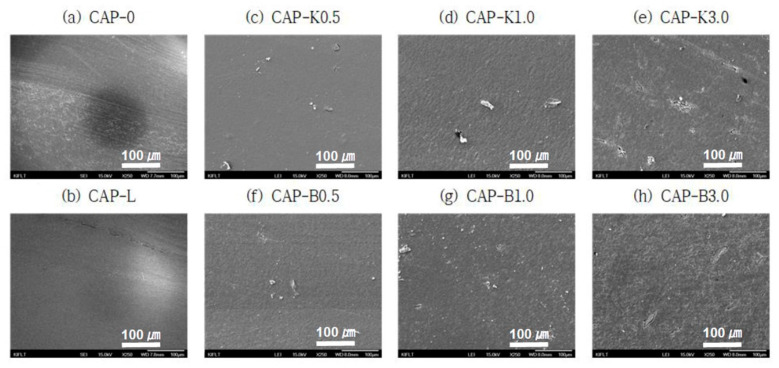
SEM images showing dispersibility of composites after adding biofillers (×250 magnification): (**a**) CAP-0; (**b**) CAP-L; (**c**) CAP-K0.5; (**d**) CAP-K1.0; (**e**) CAP-K3.0; (**f**) CAP-B0.5; (**g**) CAP-B1.0; and (**h**) CAP-B3.0.

**Figure 15 polymers-14-04286-f015:**
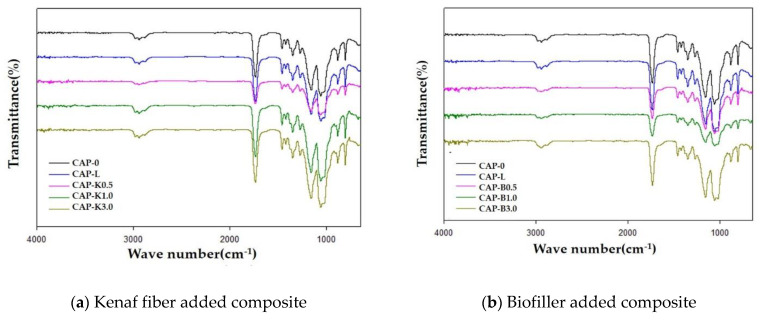
FT-IR spectra of plasticized cellulose acetate propionate–biofiller composites.

**Figure 16 polymers-14-04286-f016:**
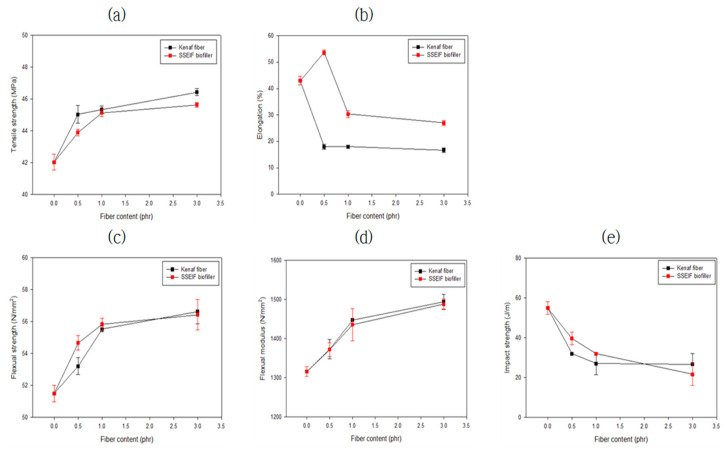
Mechanical strength of CAP–biofiller composites: (**a**) tensile strength @ break; (**b**) elongation; (**c**) flexural strength; (**d**) flexural modulus; and (**e**) impact strength (Izod, Norch).

**Figure 17 polymers-14-04286-f017:**
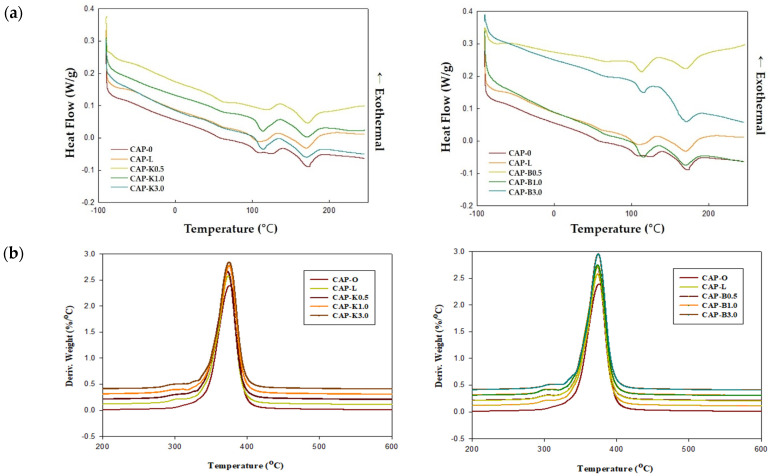
Thermal properties of cellulose acetate propionate–biofiller composites: (**a**) DSC thermograms and (**b**) DTG (derivative thermos gravimetry) curves.

**Table 1 polymers-14-04286-t001:** Weight loss according to refining (or scouring) time.

Refining Agent	Weight Loss According to Refining (or Scouring) Time					
	2 h @60 °C		4 h @60 °C		6 h @60 °C	
3%-NaOH aqueous solution	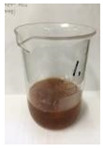	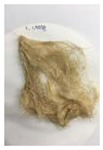	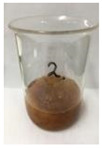	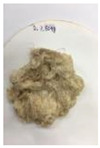	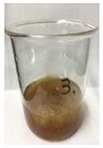	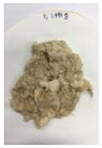
	40.94%		45.22%		46.13%	
3%-NaOH aqueous solution + α-amylase 2wt%	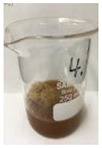	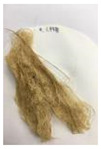	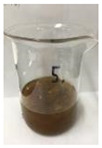	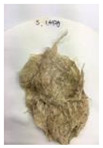	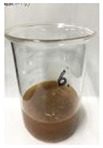	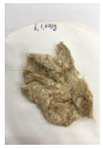
	41.38%		44.9%		48.78%	
3%-NaOH aqueous solution + α-amylase 4wt%	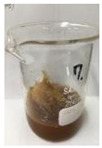	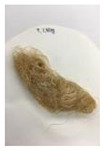	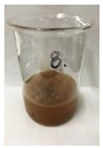	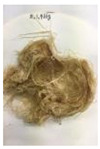	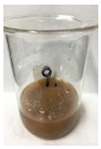	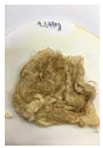
	44.79%		43.18%		43.72%	
6%-NaOH aqueous solution	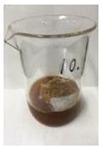	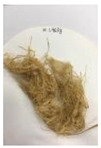	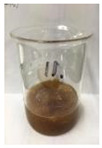	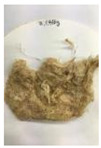	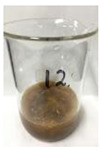	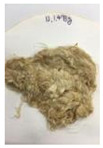
	44.6%		47.12%		47.73%	
6%-NaOH aqueous solution + α-amylase 2wt%	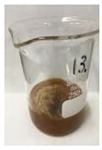	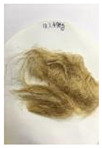	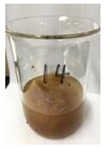	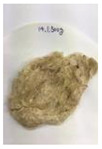	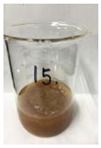	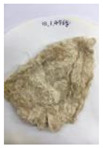
	41.44%		44.81%		43.86%	
6%-NaOH aqueous solution + α-amylase 4wt%	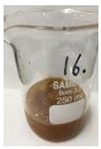	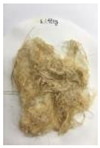	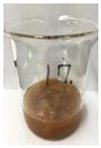	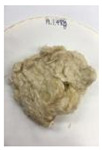	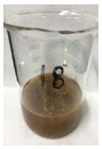	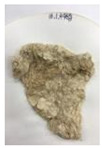
	41.47%		43.27%		44.57%	

**Table 2 polymers-14-04286-t002:** Preparation of CAP–biofiller composite.

Sample Code	Materials					
	CAP482-20	PEG300	Adipic Acid Polyester	Lactic Acid	Kenaf Fiber	SSEIF Biofiller
CAP-0	100	5	5	-	-	
CAP-L	100	5	5	2.5	-	-
CAP-K0.5	100	5	5	2.5	0.5	-
CAP-K1.0	100	5	5	2.5	1.0	-
CAP-K3.0	100	5	5	2.5	3.0	-
CAP-B0.5	100	5	5	2.5	-	0.5
CAP-B1.0	100	5	5	2.5	-	1.0
CAP-B3.0	100	5	5	2.5	-	3.0

**Table 3 polymers-14-04286-t003:** Mechanical properties of CAP–biofiller composites with different contents.

No.	CAP/PEG300/Adipic Acid Polyester/Lactic Acid/Biofiller (phr)	Hardness (D type)	Density (g/cc)	Tensile Strength (MPa)	Elongation (%)	Flexural Strength (MPa)	Flexural Modulus (MPa)	Impact Strength (J/m)
CAP-0	100/5/5/0/0	72	1.229	47.7	45	59.5	1510	43
CAP-L	100/5/5/2.5/0	70	1.231	42.9	43	51.5	1316	55
CAP-K0.5	100/5/5/2.5/K0.5	70	1.231	45.0	18	53.2	1373	32
CAP-K1.0	100/5/5/2.5/K1.0	69	1.230	45.3	18	55.5	1448	27
CAP-K3.0	100/5/5/2.5/K3.0	68	1.230	46.4	17	56.6	1494	27
CAP-B0.5	100/5/5/2.5/B0.5	71	1.228	43.7	54	54.7	1372	32
CAP-B1.0	100/5/5/2.5/B1.0	72	1.228	45.1	30	56.4	1436	32
CAP-B3.0	100/5/5/2.5/B3.0	75	1.232	45.6	27	55.8	1488	22

**Table 4 polymers-14-04286-t004:** The compositions and thermal properties of the CAP–biofiller composites.

No.	CAP/PEG300/Adipic Acid Polyester/Lactic Acid/BIOFILLER (phr)	Biofiller (phr)	Tg (°C)	Tm (°C)/ΔH(J/g)	Tm (°C)/ΔH(J/g)	Td * (°C)
CAP-0	100/5/5/0/0	0	−45.01/50.04	107.79/4.984	171.73/5.945	375.41
CAP-L	100/5/5/2.5/0	0	−43.64/55.69	108.35/5.510	170.01/7.164	374.40
CAP-K0.5	100/5/5/2.5/K0.5	K0.5	−38.01/53.84	118.14/3.409	169.87/6.243	372.38
CAP-K1.0	100/5/5/2.5/K1.0	K1.0	−47.73/51.32	113.34/5.496	168.47/5.826	375.35
CAP-K3.0	100/5/5/2.5/K3.0	K3.0	−44.28/50.56	113.22/4.739	167.09/5.761	374.40
CAP-B0.5	100/5/5/2.5/B0.5	B0.5	−38.84/53.46	113.54/3.935	170.71/6.141	373.39
CAP-B1.0	100/5/5/2.5/B1.0	B1.0	−43.57/57.79	114.41/5.200	167.74/5.971	373.39
CAP-B3.0	100/5/5/2.5/B3.0	B3.0	−42.83/53.08	113.34/2.642	164.68/8.213	374.40

* Decomposition temperatures measured by TGA for 2.5% weight loss.

## Data Availability

Not applicable.

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
