# Peer review of "Interfacial Properties and Melt Processability of Cellulose Acetate Propionate Composites by Melt Blending of Biofillers"

_polymers, 2022, doi:10.3390/polym14204286_

Round 1

Reviewer 1 Report

In the present study, a biocomposite was prepared by melt-mixing natural fibers using a cellulose acetate derivative as a polymer matrix. The natural fiber was a product with increased interfacial bonding with the substrate by surface treatment through a scouring process. The experiment of an attempt was made to remove impurities on the surface through a refining process of natural fibers and to increase compatibility with the polymer matrix. The results indicate that interfacial properties can be increased through interfacial interactions between two different components due to appropriate surface treatment.

The work may be beneficial to many industrial areas, such as material preparation, mechanical engineering and aeronautics. The logic of the article is well organized. The language is very good, -there are nearly no grammar mistakes. The work can be considered for publication after some modifications.

1. The so-called interfacial force should be correctly named as “interfacial energy” or “work of adhesion”. Please refer to the related work: Liu et al., A unified analysis of a micro-beam, droplet and CNT ring adhered on a substrate: Calculation of variation with movable boundaries. Acta Mechanica Sinica, 2013, 29(1): 62–72.

2. The citation of references should be in a correct form, such as “[5].”, and not “.[5]”.

3. The authors mentioned that, “…that can improve interfacial adhesion through mechanical bonding by increasing the roughness and surface area of the fiber…”. This indicates that the wettability or adhesion capability of the surfaces is correlated with the surface roughness and surface chemistry. This has been quantitatively illustrated in a recent work: Yu et al., Wettability enhancement of hydrophobic artificial sandstones by using the pulsed microwave plasma jet. Colloid and Interface Science Communications, 2020, 36: 100266.

4. What is the difference between the tensile strength and flexural strength? In theory, they do not have any difference.

5. In L154 of P6, the unit of the pressure is not right, and it should be “kgf/cm2”.

6. There are no equations or formulas in the present work. It is suggested to perform some quantitative studies based on continuum mechanics in future.

Reviewer 2 Report

Manuscript ID polymers-1899093

The present manuscript reports the preparation of composites based on cellulose acetate propionate (CAP). Even though the reported subject is of interest, the way the manuscript is presented is not suitable for a scientific publication in a high-impact journal.

The way the authors prepared the document presented is very poor. There is not possible to follow the ideas along with the text, and many results are discussed without any support, thus completely speculation. Thus, I regret that my recommendation is to reject the manuscript.

In the introduction, the authors presented several schemes. Schemes are a good way to summarize information and facilitate understanding. However, the schemes presented (mainly schemes 2, 3 and 4) do not add anything to the text, and thus should be removed or changed to add information to the manuscript.

The physic-chemical behaviour of cellulose-based only in hydrogen bonding is not the clearest picture. Please see works such as https://doi.org/10.1017/S0033583521000019.

The results discussion needs to be deeply improved. The authors talk about the removal of hemicellulose, lignin and wax, but did not present any technique that can prove this. Thus, all this discussion is purely speculative. At least, the authors should provide the chemical composition of the samples before and after the purification method, based for example on the TAPPI methodology.

About the purification method, the authors did not give any information about the use of enzymes during the materials and methods section. If the enzyme is the “surface treatment agent” should be described in detail and wasn’t. The same for the “scouring” which appears in the results but no details are given in the methods section.

What do the authors mean by VOC generated from natural fibres? Are volatile organic compounds? Is expected the generation of VOCs in a composite production?

Lines 178-182: It is not possible to understand what the authors want to explain.

Lines 193-196: Authors discuss the crystallinity of the composites. Based on what? None of the techniques presented in the manuscript is suitable to measure crystallinity. The same for the molecular weight. Which means “crystallinity of the crystal”?

FTIR is not discussed. Only a sentence is presented saying that no differences can be found. Thus, if it is not possible to get relevant information from FTIR I suggest removing it.

The conclusions section should be improved, and the text carefully verified.

Reviewer 3 Report

In this study, a biocomposite was prepared by melt-mixing natural fibers using a cellulose acetate derivative as a polymer matrix. The whole study is very systematic. But the manuscript must minor revise before publish.

The abstract part: I think the author should give more information about experiment data.

Figure 13. The scale is ambiguous.

Table 2. It should be given in the form of a table, not a picture.

Figure 15. The error bar is required.

Fig. 16a, Please mark the direction of the exothermal.Fig. 16a, Please mark the direction of the exothermal.

For clarity, Fig. 16b should be replaced with a DTG curve

Reviewer 4 Report

1. In the introduction section the optimum results be introduced.

2. The figure. 1 is not mentioned into the text.

3. Referring in form of [7-17] is not proper. Be modified and the unnecessary references be removed.

4. The caption of Figure 12 be corrected. It is not microscopic images.

5. How many samples were tested for reporting each results? Be mentioned in the text.

6. The best data and results be mentioned in the conclusion section.

7. Referring in results and discussion section is not proper. More references be used for your discussions.

Round 2

Reviewer 2 Report

Manuscript ID polymers-1899093R1

 The changes made by the authors do not clarify the questions raised previously. The attempts of the authors to justify some parts of the methodologies and results only introduce more mistakes and confusion. For example, the authors talk about pectin solubilization. Kenaf fibres, to the best of my knowledge, do not have pectin in their composition. NaOH is not a drug. A chemical perhaps. The cellulose and cellulose derivatives are still not understood by the authors. Thus, my recommendation still is the rejection of the manuscript.

Round 3

Reviewer 2 Report

After two rounds of revisions, some questions are still without an acceptable reply. I do not think that more revisions will solve the problem. Perhaps the reviewer and the authors are not in syntony about the comments. Thus, in my opinion, a decision should be made by the academic editor to conclude this manuscript review process.